# Oxidative Stress Reaction to Hypobaric–Hyperoxic Civilian Flight Conditions

**DOI:** 10.3390/biom14040481

**Published:** 2024-04-15

**Authors:** Nikolaus C. Netzer, Heidelinde Jaekel, Roland Popp, Johanna M. Gostner, Michael Decker, Frederik Eisendle, Rachel Turner, Petra Netzer, Carsten Patzelt, Christian Steurer, Marco Cavalli, Florian Forstner, Stephan Pramsohler

**Affiliations:** 1Institute of Mountain Emergency Medicine, Eurac Research, Noi Park Campus, Via Hypatia 2, 39100 Bozen, Italy; frederik.eisele@eurac.edu (F.E.); rachel.turner@eurac.edu (R.T.); 2Hermann Buhl Institute for Hypoxia and Sleep Medicine Research, Department Psychology and Sport Science, University Innsbruck, 6020 Innsbruck, Austria; petra.netzer@gmx.net (P.N.); s.pramsohler@gmx.net (S.P.); 3Division Sports Medicine and Rehabilitation, Department Internal Medicine, University Hospitals, 89070 Ulm, Germany; 4Terra X Cube, Eurac Research, 39100 Bozen, Italy; carsten.patzelt@eurac.edu (C.P.); christian.steurer@eurac.edu (C.S.); florian.forstner@eurac.edu (F.F.); 5Institute of Medical Biochemistry, Medical University of Innsbruck, 6020 Innsbruck, Austria; heidelinde.jaekel@i-med.ac.at (H.J.); johanna.gostner@i-med.ac.at (J.M.G.); 6Sleep Medicine Work Group, Department Psychiatry and Psychotherapy, University Hospitals, University Regensburg, 93053 Regensburg, Germany; roland.popp@medbo.de; 7Institute for Aerospace Physiology, Department Physiology, Medical School, Case Western Reserve University, Cleveland, OH 44120, USA; mjd6@case.edu

**Keywords:** high altitude, hyperoxia, flight conditions, oxidative stress, glutathione-peroxidase

## Abstract

Background: In military flight operations, during flights, fighter pilots constantly work under hyperoxic breathing conditions with supplemental oxygen in varying hypobaric environments. These conditions are suspected to cause oxidative stress to neuronal organ tissues. For civilian flight operations, the Federal Aviation Administration (FAA) also recommends supplemental oxygen for flying under hypobaric conditions equivalent to higher than 3048 m altitude, and has made it mandatory for conditions equivalent to more than 3657 m altitude. Aim: We hypothesized that hypobaric–hyperoxic civilian commercial and private flight conditions with supplemental oxygen in a flight simulation in a hypobaric chamber at 2500 m and 4500 m equivalent altitude would cause significant oxidative stress in healthy individuals. Methods: Twelve healthy, COVID-19-vaccinated (third portion of vaccination 15 months before study onset) subjects (six male, six female, mean age 35.7 years) from a larger cohort were selected to perform a 3 h flight simulation in a hypobaric chamber with increasing supplemental oxygen levels (35%, 50%, 60%, and 100% fraction of inspired oxygen, FiO_2_, via venturi valve-equipped face mask), switching back and forth between simulated altitudes of 2500 m and 4500 m. Arterial blood pressure and oxygen saturation were constantly measured via radial catheter and blood samples for blood gases taken from the catheter at each altitude and oxygen level. Additional blood samples from the arterial catheter at baseline and 60% oxygen at both altitudes were centrifuged inside the chamber and the serum was frozen instantly at −21 °C for later analysis of the oxidative stress markers malondialdehyde low-density lipoprotein (M-LDL) and glutathione-peroxidase 1 (GPX1) via the ELISA test. Results: Eleven subjects finished the study without adverse events. Whereas the partial pressure of oxygen (PO_2_) levels increased in the mean with increasing oxygen levels from baseline 96.2 mm mercury (mmHg) to 160.9 mmHg at 2500 m altitude and 60% FiO_2_ and 113.2 mmHg at 4500 m altitude and 60% FiO_2_, there was no significant increase in both oxidative markers from baseline to 60% FiO_2_ at these simulated altitudes. Some individuals had a slight increase, whereas some showed no increase at all or even a slight decrease. A moderate correlation (Pearson correlation coefficient 0.55) existed between subject age and glutathione peroxidase levels at 60% FiO_2_ at 4500 m altitude. Conclusion: Supplemental oxygen of 60% FiO_2_ in a flight simulation, compared to flying in cabin pressure levels equivalent to 2500 m–4500 m altitude, does not lead to a significant increase or decrease in the oxidative stress markers M-LDL and GPX1 in the serum of arterial blood.

## 1. Introduction

In flight operations with fighter jets and in space crafts, inspired levels of hypoxic and hyperoxic air for pilots and astronauts vary, largely due to the instability of the oxygen delivery system and pressure losses in the flight cabin. This can have an influence on the blood flow and oxygenation of the brain and on cognitive performance [1]. The change of oxygenation of arterial blood can also change peripheral arterial blood pressure and blood flow [2]. The rapid changes in oxygenation of neuronal cells and the oxidative stress can lead to the production of inflammatory neuronal cell markers [3]. Hyperoxia is suspected to cause brain damage and possible white matter densities, proven at least in animal research [4,5]. However, in humans also, hyperoxic flight operations for space flights showed microstructural changes in neuronal tissues as a predisposition for white matter densities [6].

Additionally, in humans, repeated hypobaric hyperoxia increases inflammatory markers and might lead to symptoms of chronic fatigue [7].

Previous investigations with different FiO_2_ (fraction of inspired oxygen) levels of supplemental oxygen in simulated flight operations showed a dose-dependent response to higher FiO_2_ levels concerning cerebral blood flow and brain oxygenation; a statistical difference in response could be seen at FiO_2_ levels from 60% on [8]. Despite this reduction in blood flow, it seems that higher oxygenation enhances cognitive function and high-density EEG (electroencephalogram) activity. For the in-flight performance of flight crews, supplemental oxygen and high FiO_2_ could be an advantage [9]. This is the main reason why not only is supplemental oxygen standard in military flight operations, but also, the United States Federal Aviation Administration (FAA) recommends supplemental oxygen in non-pressurized civilian airplanes at altitudes from 3750 m and makes it mandatory from altitudes of 4200 m [10]. Even in pressurized airplanes, supplemental oxygen must be ready for pilots and flight crews [11]. 

Magnetic resonance imaging (MRI) scans have shown in studies with hypoxic conditions in humans a reduction of cerebral blood flow in several brain regions, although from a physiologic standpoint, one might assume that hypoxia increases blood flow and hyperoxia decreases blood flow [12].

Hypoxia-related accidents in military flight operations have been recorded and published for a long period of time [13,14,15], but they also affect civilian flight operations and, according to post-accident investigations, have led to major crashes like the one with famous golfer Payne Stewart on board, as well as to a Greek airplane crash with 121 fatalities [16,17]. Therefore, supplemental oxygen for flight operations makes sense not only in military flights, but also for all civilian flight operations. Some researchers even lobby a reduction of hypoxia in civilian flights for crew and passengers from an altitude level of 2400 m in the pressurized cabin down to 1500 m equivalent altitude [18].

Supplemental oxygen in civilian flight operations for flight crews is usually delivered at FiO_2_ levels of 35–60% via face masks. The question is if supplemental oxygen and the achievement of a hyperoxic arterial blood gas level lead to a possibly harmful level of oxidative stress for neuronal cells. 

Therefore, we aimed to investigate, in a crossover physiology study with a flight simulation with healthy humans in a hypobaric chamber at the most common cabin altitude levels in civilian flight operations, if the delivery of supplemental oxygen leads to a significant increase of the oxidative stress markers malondialdehyde-modified low-density lipoprotein (M-LDL) and glutathione peroxidase 1 (GPX1) in arterial blood. We hypothesized that supplemental oxygen with an FiO_2_ level of 60% would lead to a significant increase of the two oxidative stress markers versus baseline.

## 2. Methods

### 2.1. Subjects

Subjects were recruited via social media and Blackboard announcement. The inclusion criteria were an age between 20 and 55 years, an overall completely healthy physical status without any need for medications, non-smoking, non-pregnancy, and a COVID-19 vaccination status with three completed COVID-19 vaccinations at least 15 months before the study onset. 

The initial cohort from this recruitment consisted of 50 volunteers, of which 12 males and 12 females were willing to participate and were scheduled for this study. All 24 persons underwent an extensive clinical exam by an internal medicine physician, including a lung function test (NDD Easy One, NDD Zuerich, Zuerich, Switzerland) to exclude any ongoing respiratory disease or actual infection. All subjects had an FVC (forced vital capacity) and FEV1 (forced expiratory volume in one second) > 100% of the normal values based on their age and body size.

Via coin tossing, while keeping an equal sex distribution, the 24 subjects were divided into two groups. Group one performed the flight simulation with a radial catheter and group two performed the flight simulation parallel without a radial catheter for blood sampling, standing by as reserve in case of problems with the puncture of the arteria radialis when placing the radial catheter in a subject of group one. The mean age of the 12 subjects from group one was 35.7 years and the standard deviation (SD) was 12.35 years.

The study protocol, according to the standards of the Declaration of Helsinki including amendments, and the subject information were reviewed and accepted by the ethical review board of the province of Bozen/South Tyrol (106-2020) and also by the ethical review board of the University of Innsbruck, Austria (42-2019).

### 2.2. Study Site and Flight Simulation

Terra X Cube is one of the world’s largest civilian hypobaric chambers, with a size of 137 m^2^ and a room height of 6 m (822 cubic meter volume). It is located in Bozen, Italy, at an altitude of 322 m above sea level and is incorporated in Eurac Research. The maximal reachable altitude is 9000 m; the maximal rise in altitude is normally 6 m/s. For the Hypoxiflight study, we stretched the rising speed to 10 m/s and used a speed of 50 m/s for the decrease of altitude. The room temperature was kept at a comfortable 22 °C at all times. One subject from group one and one subject from standby group two were in the chamber at the same time for three hours. The study was performed on six consecutive days with two shifts per day: one from 9–12 a.m. (9.00–12.00) and one from 2–5 p.m. (14.00–17.00).

The radial catheter (BD Flow-Switch, Becton Dickinson, Heidelberg, Germany) was placed in the arteria radialis of the non-prominent arm of the subject in a supine position under ultrasound control by an anesthesiologist before the start of the trial and was connected to a patient monitor (Phillips IntelliVue, Phillips Healthcare, Amsterdam, The Netherlands). The arterial blood pressure (BP), arterial oxygen saturation (SaO_2_), and pulse frequency were constantly recorded from that time point on. Baseline arterial blood samples (one for blood gases, one for centrifugation to gain serum) were drawn from a three-way valve before the first venturi valve-equipped face mask was put on the subject with 35% FiO_2_ and 8 L/min O_2_ flow.

Since PO_2_ (partial oxygen pressure) analyzed from arterial blood gases is considered the gold standard for the measurement of arterial blood oxygenation, we concentrated on this parameter for the assessment of the oxidative level of the subjects at the different steps. Blood gas analysis was performed via a Roche Opti cartridge system (Opti CCA TS, Opti Medical, Atlanta, GA, USA) on site inside the chamber. The analyzer was calibrated for the specific altitude each day at each step.

After baseline measurements, the altitude was increased and decreased back and forth between 2500 m and 4500 m, respectively, and the equivalent barometric pressure was achieved according to a specific protocol mimicking a standard civilian flight operation (for example, a flight over a mountain range like the Alps or the Rocky Mountains) or after cabin pressure loss in a pressurized airline jet. The flight simulation protocol is shown in Figure 1. After each return from 4500 m to 2500 m altitude, respectively, with the equivalent air pressure in the Terra X Cube, the face masks were changed on the subject with the specific venturi valves for the FiO_2_ steps of 35%, 50%, 60%, and 100%. Blood gas analysis was performed at each FiO_2_ step and at each altitude of the protocol.

### 2.3. Assessment of Oxidative Stress

According to the leading literature, we have chosen malondialdehyde-modified low-density lipoprotein (M-LDL) and glutathione peroxidase 1 (GPX1) as oxidative stress markers for our analysis, and the proof of the hypothesis that there is a correlation between arterial hyperoxia and oxidative stress in the serum from arterial blood in our human subjects [19,20,21,22]. According to the German Ministry of Science, these are also the two markers that express oxidative stress the best on a feasible level of cost and analysis efforts [23].

Additional arterial blood for centrifugation inside the hypobaric chamber was drawn from the radial catheter at baseline and at FiO_2_ of 60% at both altitudes. The serum samples were transported outside the chamber via an airlock immediately after centrifugation and frozen at −21 °C for further analysis at the biochemistry laboratory of the Medical University of Innsbruck. For the transport from Bozen to Innsbruck (driving time 1 h 15 min), after the collection of all serum samples at the end of the study, the serum was kept in a Styrofoam box with dry ice. In the biochemistry laboratory, the biomarkers oxidized-LDL/MDA adducts and GPX1 were analyzed via ELISA, ox-LDL/MDA adduct ELISA K7810 (Immundiagnostik, Bensheim, Germany), and Human Glutathione Peroxidase 1 ELISA ab193767 (Abcam plc, Cambridge, UK). Each test was repeated twice to prove reproducibility.

### 2.4. Statistical Analyses

An a priori power analysis was performed based on our hypothesis with an estimated high correlation and a Pearson correlation coefficient of 0.8 between PO_2_ values and oxidative stress biomarker ELISA results. At a significance level of *p* 0.05 (5%) and a power of 0.8 (80%), the necessary number of subjects was nine [24].

All values here are presented in means and standard deviation. The significance of difference between two values was calculated via Student’s *t*-test for paired variables; a significant difference between variables is presented as *p* ≤ 0.05. The correlation between parameters was calculated according to Pearson and expressed with the Pearson correlation coefficient. All statistical work was performed with the Excel 2021 (Microsoft, Seattle, WA, USA) statistics program. 

## 3. Results

Eleven of the twelve selected subjects and the twelve standby subjects finished the flight simulation in the hypobaric chamber according to the stepwise altitude increase and decrease and supplemental oxygen increase without any problems. One subject (female, 48 years) dropped out after baseline and 35% FiO_2_ supplemental oxygen at simulated 2500 m due to subjectively felt stress breathing under the face mask and subsequent hyperventilation, leading to dizziness. After taking the facemask off, she calmed down to a completely normal respiration and all feelings of dizziness disappeared, but she did not want to continue with the protocol.

Arterial PO_2_ measured from the blood gases increased from baseline (322 m altitude, 9761 kPa air pressure) in the mean 96.2 mmHg (SD 9.9 mmHg) to 160.9 mmHg (mean, SD 20.5 mmHg) at 2500 m simulated altitude with 60% FiO_2_. At the equivalent 4500 m and 60% FiO_2_, the mean PO_2_ was at 113.2 mmHg (SD 5.7 mmHg). All individual PO_2_ values from baseline to 35%, 50%, 60%, and 100% FiO_2_ at 2500 m and 4500 m simulated altitude are presented in Figure 2.

Mean malondialdehyde-modified LDL (M-LDL) increased in the mean from 49.0 ng/mL (SD 58.02 ng/mL) at baseline to 65.5 ng/mL (SD 69.5 ng/mL) at 2500 m 60% FiO_2_ and 73.2 ng/mL (SD 99.2 ng/mL) at 4500 m 60% FiO_2_. One subject (age 38) started with a much higher base value of M-LDL (189.9 ng/mL, reason unknown; the subject had no signs of infection and lung function better than the suggested normal values) and further increased to 222.2 ng/mL at 2500 m 60% FiO_2_ and 310.9 ng/mL at 4500 m 60% FiO_2_. If this subject had been left out from the calculation, the mean M-LDL for the other 10 subjects would have fallen between 2500 m 60% FiO_2_ and 4500 m 60% FiO_2_ from 43.1 ng/mL to 39.3 ng/mL with much lower standard deviations. With and without this subject, the difference between the baseline value and value at 2500 m 60%FiO_2_ or 4500 m 60% FiO_2_ did not reach significance (n.s.). Figure 3 presents the M-LDL means and SD at the different protocol time points of measure. 

Glutathione peroxidase 1 (GPX1) in the mean decreased from baseline 33.5 g/mL (SD 29.5 ng/mL) to 25.9 ng/mL (SD 20.3 ng/mL) at 2500 m 60% FiO_2_ and also slightly from baseline to 29.6 ng/mL (SD 25.9 ng/mL) at 4500 m 60% FiO_2_. Likewise, with M-LDL, there was no significant difference between GPX1 at baseline and one of the altitudes with 60% supplemental O_2_ or in the different values between altitudes. Figure 4 shows the single GPX1 values for each subject.

There was no correlation between M-LDL and arterial PO_2_ values at baseline or at one of the altitudes with supplemental 60% oxygen. The Pearson correlation coefficient for M-LDL at 2500 m and 60% FiO_2_ was 0.02 and at 4500 m and 60% FiO_2_ 0.11. For GPX1, there was a slight negative correlation with PO_2_ values at 2500 m and 60% FiO_2_ (the Pearson correlation coefficient was −30.0) and a moderate negative correlation with PO_2_ values at 4500 m and 60% FiO_2_, with a Peason correlation coefficient of −0.38. Figure 5 shows the correlation curve for GPX1 and arterial PO_2_ at 4500 m and 60% FiO_2_.

There is a moderate correlation between subject age and GPX1 at 4500 m and 60% FiO_2_, with a correlation coefficient of 0.55.

## 4. Discussion 

To the best of our knowledge, this is the first hypobaric–hyperoxic flight simulation with specific commonly occurring altitudes and recommended maximal supplemental oxygen levels for civilian airflight operations. Our results show that the recommended oxidative stress biomarkers [19,20,21,22,23] malondialdehyde-modified low-density lipoprotein (M-LDL) and glutathione peroxidase 1 (GPX1) do not significantly increase in a simulated three-hour flight with changing altitudes changing the barometric pressure in a hypobaric chamber. This leads to the assumption that a several-hours flight with supplemental oxygen through a face mask might not cause a higher level of oxidative stress resulting in neuronal cell damage.

The baseline levels of the two measured oxidative stress biomarkers in the serum from arterial blood showed a high inter-individual variation, which led to large standard deviations from the mean values, and they did not change much in the single subjects with the altitude at 60% FiO_2_. We do not know why the baseline levels were so individually different in our healthy subject population with clinically excluded ongoing infection or major stress factors. Age did not play a major role here, at least not concerning the baseline values. However, at 4500 m and 60% supplemental oxygen, there was a moderate correlation between age and GPX1 values. The higher the age, the higher the level of this oxidative stress biomarker.

Our results confirm, in some way, the outcomes of former studies concerning military flight operations from members of our work group [7], who showed that flight operators with higher inflammatory biomarkers have a higher tendency to develop signs of chronic fatigue, but that these higher levels of inflammatory biomarkers cannot be, or are only in part, causally related to the number of performed flights and level of hypobaric hyperoxia. In fighter pilots in hypobaric hypoxic conditions, it seems that glutathione and three other oxidative stress markers increased with a higher tendency of oxidative stress elevation in experienced fighter pilots with more flight hours [25]. This could lead to two assumptions. One is that oxidative stress adds up over a longer period of flight hours, in accordance with results that showed more inflammatory biomarkers and greater chronic fatigue with total flight time in our previous studies [7]. The other is that hyperoxia might indeed suppress oxidative stress to a certain level.

With a non-significant increase of M-LDL and GPX1 at lower and higher altitudes at hypobaric air pressure, respectively, our results show that supplemental oxygen up to 60% FiO_2_ is safe under the aspect of the non-development of oxidative stress, although there is a moderate correlation between arterial PO_2_ levels and GPX1 values. The positive aspects of supplemental oxygen, as recommended by authorities for flight operations in airplanes without pressurized cabins or pressure loss in normally pressurized cabins, seem to make sense, especially since most studies show a significant cognitive decline of pilots in hypoxia from 4750 m on [26].

Other studies or position papers, mainly from high-altitude researcher John West, concerning persons hiking in high altitudes or living or working at high altitudes, have also shown that the negative aspects of hypoxia like acute mountain sickness or reduced levels of cognitive function can be reduced or eliminated with supplemental oxygen [27,28,29].

If the oxidative stress from supplemental oxygen in flight operations is unlikely the cause of neuronal cell damage and white matter densities occurring in pilots, especially fighter pilots with crews and astronauts, the question remains of what the possible causes could be. Older and newer discussions circle around the effect of high G-forces on fighter airplane crews and in astronauts repeatedly occurring during training and flight operations [30,31,32]. Another hypothesis is that weightlessness, as it occurs not only in space flight, but also repeatedly in dog fighting in military flight operations with new fighter jets, could be the cause for neuronal damages [33]. However, these discussed causes happen extremely rarely in civilian flight operations with commercial airlines or private or transport flights.

An interesting side aspect of the outcome of our study is the reason for the drop out of one subject. Hyperventilation and the consequences of hypocapnia and alkalosis with dizziness and possible tetany (paw position) due to stressful inflight situations with the necessity of supplemental oxygen might be a bigger problem in sensible passengers or flight crew members than high concentrations of inspired oxygen or subsequently high arterial PO_2_ levels. According to several publications, hyperventilation, especially in hypoxic situations, occurs not only in the passengers of commercial flights, but also in fighter pilots and professional commercial airline pilots [34,35,36]. Therefore, flight crews and pilots should be trained in breathing techniques, which they can use for themselves and also to help passengers in need.

Our study has a few limitations. One is, of course, the relatively low number of subjects, which is due to the large expense in such a hypobaric chamber study including a constant trial-placed radial line. However, although we were not able to confirm our hypothesis that oxidative stress and arterial PO_2_ levels correlate, we could show with the means and in the single subjects that there is no significant increase of the oxidative stress biomarkers even with these high levels of arterial PO_2_ in each subject in this two-to-three-hour period. Another limitation is that for feasibility and cost reasons, we could only analyze two oxidative stress biomarkers at baseline and two other points of measurement. However, we think that these two oxidative stress biomarkers and the level of 60% inspired oxygen are representative of a real civilian flight situation where supplemental oxygen is asked for.

## 5. Conclusions

Supplemental oxygen up to 60% FiO_2_ via a venturi valve-equipped face mask in a civilian flight simulation with a three-hour duration at a barometric pressure equivalent to altitudes of 2500 m and 4500 m does not lead to an increase in the oxidative stress markers malondialdehyde-modified low-density lipoprotein and glutathione peroxidase 1 in the arterial blood of healthy subjects despite high arterial PO_2_ levels. The recommended supplemental oxygen for flight crews in non-pressurized airplane cabins or in situations with pressure loss does not cause a high level of oxidative stress and is safe. 

## Figures and Tables

**Figure 1 biomolecules-14-00481-f001:**
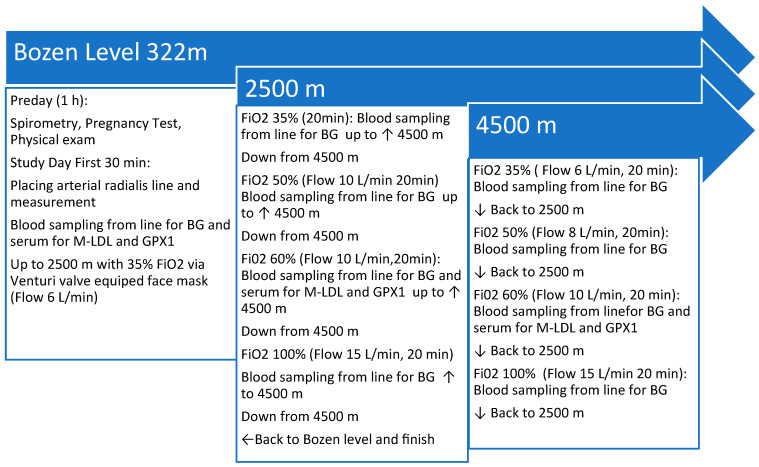
Study protocol time course with altitudes, supplemental oxygen levels, and blood sampling. FiO_2_ = fraction of inspired oxygen, BG = blood gases, Flow = airflow of inspired oxygen, M-LDL = malondialdehyde-modified low-density lipoprotein, GPX1 = glutathione peroxidase 1, L = liter, min = minutes.

**Figure 2 biomolecules-14-00481-f002:**
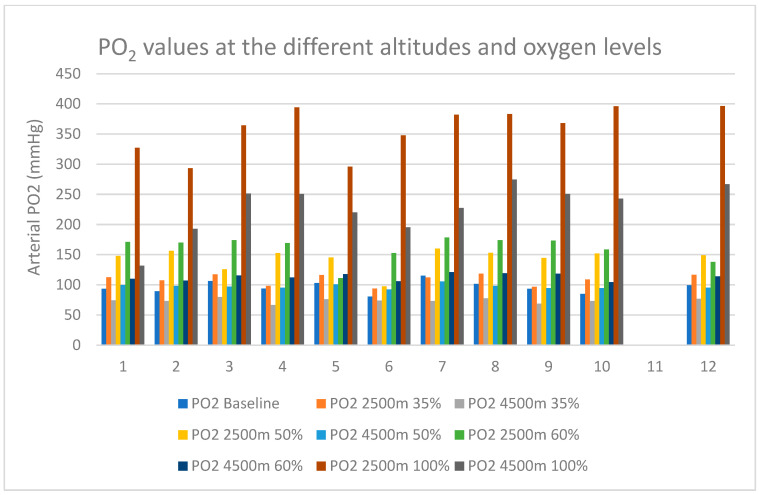
Arterial blood gas from radial line partial pressure level of oxygen in the single subjects at single points of measurement. PO_2_ = partial pressure of oxygen in arterial blood, 35–100% = fraction of inspired oxygen through venturi valve-equipped face mask at the specific hypobaric altitude level, X-Axis: subject number.

**Figure 3 biomolecules-14-00481-f003:**
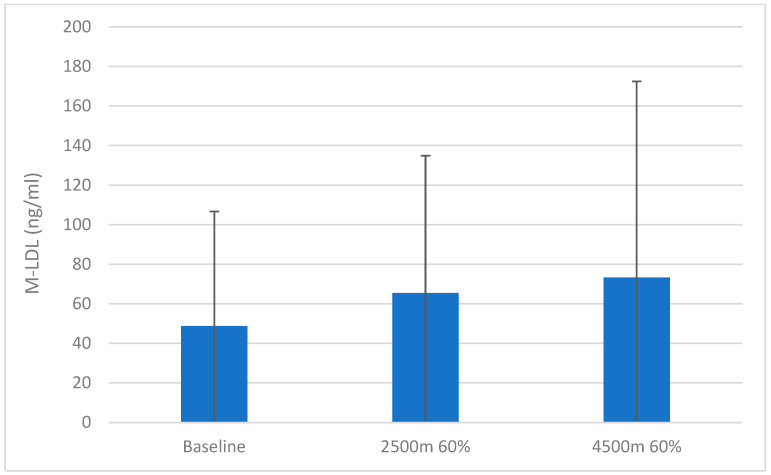
Mean values of malondialdehyde-modified low-density lipoprotein (M-LDL) at baseline and at different altitudes with 60% inspired oxygen. Data are represented as mean ± SD.

**Figure 4 biomolecules-14-00481-f004:**
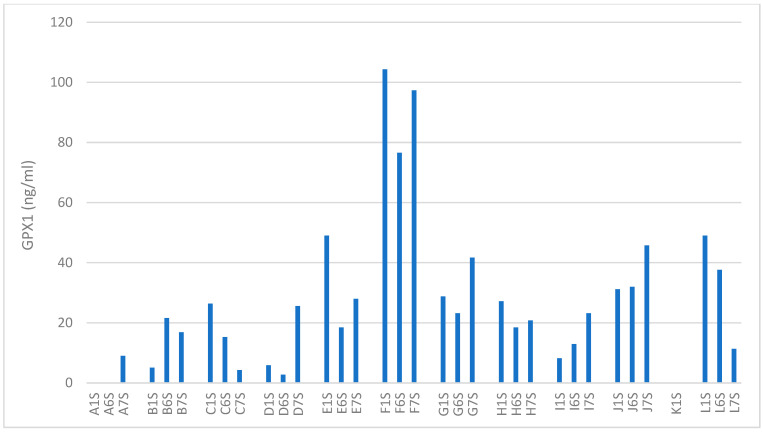
Glutathione peroxidase values of the single subjects at the different points of measurement. GPX1 = glutathione peroxidase 1, A–L = subjects (subjects are marked with letters instead of numbers; the order is the same as in Figure 2), 1S = baseline at 322 m without supplemental oxygen, 6S = 2500 m equivalent altitude with 60% FiO_2_, 7S = 4500 m equivalent altitude with 60% FiO_2_.

**Figure 5 biomolecules-14-00481-f005:**
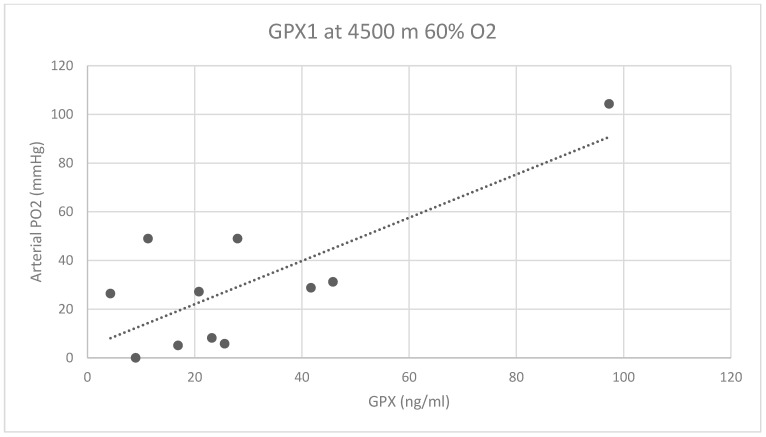
Correlation point-cloud graph of glutathione peroxidase 1 (GPX) values in correlation with arterial PO_2_ levels. 4500 m 60% = hypobaric pressure equivalent to 4500 m altitude and 60% FiO_2_ through venturi valve-equipped face mask (correlation coefficient 0.38).

## Data Availability

RAW data is available on request by the corresponding author.

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
