# Peer review of "Oxidative Stress Reaction to Hypobaric–Hyperoxic Civilian Flight Conditions"

_biomolecules, 2024, doi:10.3390/biom14040481_

Round 1
Reviewer 1 Report
Comments and Suggestions for Authors In addition, I have checked the figures and according to my opinion they are not appropriate for the journal. The authors must incorporate the respective units at x and y axes, add proper titles, and overall improve the quality of their figures.

Author Response
We thank the reviewer for his helpful and fair comments and critics and have revised the manuscript accordingly:
1. We have enriched the discussion in regard to the analysis of the two biomarkers and in regard to possible alternative reasons for oxidative stress accumulating with several flights in a row or with the problems with hyperventilation. We have added more references for these points.
2. We have now for every term written the full word in the first sentence of mention.
3. See 1. We did trust the publication from the German Ministry of Science, that if there is limited financial sources for the analysis of oxidative stress biomarkers that M-LDL and GPX 1 would be the two most sensitive and senseful markers. We also refer to the publications that glutathione peroxidase is more sensitive than glutathione directly.
4. and 5. We refer here to our remarks in 1. and 3.
6. The maximum most common FiO2 used in other trials (reference 7-9) is equally 60% , which is the highest amount delivered via simple inexpensive Venturi valve equipped mask. This is also a usual amount of oxygen in civilian flight supplemental oxygen use. 100% is usually only used in certain situation in military flight operations.
7. Misspelling Corrected
8. and 9. Since on eother reviewer wanted less graphs we made a compromise and added sentences with the other correlation coefficients from M-LDL and GPX1
10. We added references. See also 1.and 3.
Reviewer 2 Report
Comments and Suggestions for Authors
The authors of the manuscript “Oxidative Stress Reaction to Hypobaric-Hyperoxic Civilian Flight Conditions” have investigated in a simulated altitude chamber study how different oxygen doses affect oxygen saturation and stress parameters for oxidative stress. For this purpose, they were able to include 11 (?) test subjects who flew a three-hour simulated flight at altitudes of 2500 - 4500 m above sea level and who during this flight were exposed to 35%, 50%, 60% and 100% FiO2 over a Mask was additionally supplied with oxygen. At baseline and in a single 60%, oxidative stress parameters were also taken from an arterial blood sample and analyzed later. No relationship was found between oxidative stress parameters and oxygen administration.
The manuscript is well written. Even the topic is from an interest for further analysis there are several points that should be revised.
A major critical point is, that there are more authors listed on this manuscript than person examined in this study. This have to been checked.
Additional points:
Abstract: Mandatory oxygen administration is only mandatory for non-pressurized civilian airplanes. This should be clearly stated in the abstract (as well as later in the main body).
Line 32: “from a larger cohort”… What do the authors want to say here? That they selected 11 test subjects from 50 volunteers? This is unnecessary for the abstract and should be deleted.
Abstract: It should be made clearer in the abstract that the oxidative stress parameters were only recorded at two measurement times. The current impression is that this happens at significantly more measurement times (namely at baseline and four others, since four different FiO2 are administered).
Line 75f: Why are the flight levels mentioned here different to that in the abstract? This should be the same or it should be cleary explained, why there are different flight levels.
Line 77-80: The entire paragraph has nothing to do with the study and can be deleted.
Line 105: Covid-19 should be written always in the same way.
Line 117: It is not clear from the manuscript what Group 2 is dealing with and what influence this had on the results. From what it looks like to me, a group was taken on the simulated flight, but 11 of the 12 test subjects from group 1 were included in the analysis; a backup by a participant from group 2 could not be realized. It is therefore advisable to write this clearly in the methodology and to take the exclusion of one test subject into account in the average age, etc. In the end there were only 11 test subjects who were included in the analysis.
Line 159: In literature sources 19-22, the same singular author is quoted again and again. Here one of the literature sources should be used in which the original statement was made and multiple citations should be avoided.
Line 160: Is this the correct name for this Ministry? There is only a "German Ministry of Education and Research" (https://www.bmbf.de/bmbf/en/ministry/ministry_node.html)
Line 178: Was the data normally distributed? If not, mean and interquartile ranges should be reported instead of mean and standard deviation and the statistical tests should be checked.
Line 193: Make clear how many participants were included in the study…11 or 12?
Line 208: Fig. 3 with a point.
Line 230: The statement from line 230 cannot be verified without precise knowledge of the flight profile. Figure 1 is missing.
Line 263: The G-loads of pilots of high performance aircraft and astronauts have different G-loads, especially regarding the G-acceleration axis, and are therefore not comparable. But this is done here. This should be clearly defined.
Line 265f: This is a thesis that should either be supported by a reference or deleted.
Line 285: If the statement in this sentence is true, I don't understand why the study authors allowed the test subjects to breathe up to 100% FiO2. What was the point of this study approach if it was just about oxidative stress?
Line 293: In the case of rapid decompression, every aircraft would automatically lose altitude directly in an emergency procedure and then possible oxidative stress for the time until the emergency landing represents a secondary, negligible parameter. Therefore, the mention of rapid decompression is, in my opinion, important for this study negligible and can be deleted.
References 23: The correct citation is: “Robert Koch-Institut. Oxidativer Stress und Möglichkeiten seiner Messung aus umweltmedizinischer Sicht. Bundesgesundheitsbl. 51, 1464–1482 (2008). https://doi.org/10.1007/s00103-008-0720-5“
In total, the manuscript needs a major revision before it could be reviewed again.
Comments on the Quality of English LanguageN/a
Author Response
We thank the reviewer for his expertise high quality critical comments.
This is our response:
- We have removed two authors from the author list. We understand that the number of authors is large for a physiology trial. However, this trial in a large hypobaric chamber required a lot of effort from different departments and specialties. We divided the working group in three work fields (physiologic biology, psychology and cognitive function and cardiovascular and cerebral function). Nevertheless, all members of each group supported the other groups in the working process and has his input in each field, which we covered with this trial. Therefor we would like to list the authors now in the authors listing in the revision.
- We have added the reviewers recommendation in the abstract.
- We corrected the sentence with "cohort". It was a misspelling respectively wrong sentence construction.
- We added an extra sentence making clear that oxidative stress markers were only recorded at two time points in the trial.
- We have chosen other levels than the, based on feet, limit altitude levels of the FAA in order to see a clear biophysiologic effect and because these were the altitude levels chosen in comparable studies with military aircraft personal. We wanted to compare our results to the results of theses studies.
- We somewhat agree with the reviewer that standing alone this para makes no sense, but we don't want to delete it. Therefore we added a sentence to bring this para in a better context.
- Covid-19 is now written in one way throughout the manuscript.
- We changed the para with the subjects and made clear that 12 participants were in standby for a dropout of one of the subjects before! study onset. We disagree with the reviewer that the completion of only 11 subjects should be written in the method section. This is clearly a point for the result section that of 12 subjects one didn't conclude the the study because of a dropout after study onset due to a side effect.
- We disagree with the reviewer, that an author dealing with the best of possible oxidative stress markers cannot be listed with several articles, which confirm our selection of M-LDL and GPX1. Especially since another reviewer exactly wanted clear arguments for the selection of these two oxidative stress markers.
- We have made a correction of the Ministry. However the Ministry has changed it name during changes of the government. The recommendation was published when the ministry had a different name.#
- Data was normally distributed and statistics checked by a statistician from Innsbruck University.
- We changed this para in the results to make clear that 11 subjects finished the study.
- Corrected.
- We added the reviewers statement regarding the G Loads in the text.
- References have been added.
- We agree with the reviewer that the 100% oxygen does not play a role here in regard of this biophysiologic study and it did not interfere with the probes since in all subjects they were taken before the 100% O2. Since for another group of researchers in our entire study group 100% could make a difference in the cerebral blood flow subjects received 100% O2. We though we mention this although it plays no role for the study within the entire trial presented here.
- We disagree here with the reviewer. I myself as first author was in an almost crash in an Argentinian Airline airplane flying from Mendoza to Buenos Aires April 1990. The time after pressure loss at regular flying altitude until the airplane was stabilized at 1500-2000 m was at least 20 min. Although this per se might not cause enough oxidative stress, one has to take the hyperventilation of supplemental oxygen into account, which happened in the passengers under existential stress.
- We changed the reference according to the reviewers suggestion. Although we read the complete article several times, we have taken the reference from Pubmed and not from the article itself.
Reviewer 3 Report
Comments and Suggestions for Authors
The authors did an interesting study but must modify several parts of the manuscript.
The Title must be modified because as it is, gives the idea that the oxidative stress was studied in many organs and it was realized in both, vaccinated and non-vaccinated persons. For the accuracy, the title must mention the oxidative stress reactions observed in arterial blood in vaccinated persons. The same information must be found also in Abstract.
At Results section, a graph per entire group, separated by altitude and partial pressure of oxygen must be realized, even if the present graph remains in manuscript, thus the reader will see the modifications per experimental group and not per person.
It is also necessary a graph for GPX per group and altitude.
At Discussion part, the authors must not be so sure about the safety of oxygen administration since every organ acts differently and the blood has its own response at hyperoxia. Since the study was not done in parallel with experiments on animals that could give answers at the questions about the organ responses, the author must not conclude that this experiment proves the safety of hyperoxygenation with no harmful effects on neurons. In humans, different compartments respond in different manners, and in text, the real results must be noted: in arterial blood, the oxidative stress was not increased.
In Discussion part, the other factors that are involved at high altitude, during commercial flights, are not mentioned, factors that could affect the oxidative stress in organs, for example UV light or other radiation that can accelerate the cataract development producing the singlet oxygen, and so on.
It is necessary to maintain the discussion at the arterial blood level and many other factors to be mentioned as possible triggers or accelerators of oxidative stress in organs.
At Conclusions, it must be mentioned that the oxidative stress was investigated in arterial blood in vaccinated persons.
Also, must be mentioned that other studies are necessary to prove the safety of hyperoxygenation before being recommended.
A study made on 11 people is not enough for recommendation that will be applied on all pilots, vaccinated or not.
At References part, the internet pages (sites) must be replaced by scientific papers from scientific journals, if it is possible.
The references are not written in the standard style, so they must be arranged.
Comments on the Quality of English Language
In the text, some small mistakes are seen.
Author Response
- We have now written in the abstract and methods that Covid-19 vaccination was completed in every subject at least 15 month prior to study start. We do not think that the Vaccination had any effect in regard to oxidative stress at this time point. The vaccination was only mentioned since a hidden Covid- 19 infection could have influenced oxidative stress results. But we tested clinically including with the spirometry that all subjects were completely healthy without infection at study onset.
- We do not understand the reviewer in regard to his critical remarks that oxidative stress markers were only measured in arterial blood, since already in our original submission in all parts including the conclusions it was written "in arterial blood".
- We added not further graph since we got here different opinions from the three reviewers. We think the actual graphs are sufficient, but we added in the text the correlation coefficients between oxidative stress markers and arterial PO2 for all measurements.
Round 2
Reviewer 1 Report
Comments and Suggestions for Authors
Manuscript Number: biomolecules-2812437
Journal: Biomolecules (ISSN 2218-273X)
Title: Oxidative Stress Reaction to Hypobaric-Hyperoxic Civilian Flight Conditions
Authors: Nikolaus Christoph Netzer * , Heidelinde Jaeckel , Roland Popp , Johanna Maria Gostner , Michael John Decker , Frederik Eisele , Rachel Turner , Giulia Roveri , Stefanie Eckert , Petra Gertrud Netzer , Carsten Patzelt , Christian Steurer , Marco Cavalli , Florian Forstner , Stephan Pramsohler
The authors hypothesized that hypobaric hyperoxic civilian commercial and private flight conditions with supplemental oxygen in a flight simulation in a hypobaric chamber at 2500m and 4500m equivalent altitude would cause significant oxidative stress in healthy individuals. They found that supplemental oxygen of 60% FiO2 in 46 a flight simulation comparing to flying in cabin pressure levels equivalent to 2500m-4500m altitude 47 does not lead to a significant increase or decrease in the oxidative stress markers M-LDL and GLP 48 in serum of arterial blood.
The manuscript is appropriate for the journal, it provides important data and can be published with its present form with some minor changes. The writing is concise, easy to follow, and interesting, without repetition but it is also recommended to take into account advice from an English language expert because it needs some amendments in English language of the manuscript.
1. Please correct missing gaps and misspellings (examples in abstract in methods, in page 4 in third paragraph, page 5 in third paragraph, page 6 in the end of first and in the beginning of second paragraph).
2. Please use formal language (page 9 in the third paragraph, “didn’t’’).
3. Please be consistent of using commas or points in numbers.
4. Please mention the units in the main graphs in fig 3 and 5.
5. Please answer a previous question. What is the basis for your assumption that the levels of both biomarkers will increase, and they will not decrease or remain stable?
6. In page 8, please mention the importance of testing these biomarkers, adding more recent references in the first paragraph in the section of assessment of oxidative stress.
7. In page 8, you gave an answer in a previous question in the section of assessment of oxidative stress in the second paragraph. Please mention your answer in the main text and mention the references accordingly.
8. In page 11, in the first paragraph of discussion part, please add more recent references for the using of the tested biomarkers.
9. In page 11, please check the last sentence of the first paragraph of discussion part, it is too absolute.
10. I suggest you enrich the discussion section with further. Some simple proposals can be found below:
• https://doi.org/10.1177/0748233709103032
• https://doi.org/10.1080/00140139.2021.1931474
• https://doi.org/10.1080/00140139.2020.1842514
Author Response
We thank the reviewer for his critical remarks for the improvement of the text and have revised the last version accordingly to his requests.
Reviewer 3 Report
Comments and Suggestions for Authors
Since the authors refused to improve their work, the manuscript does not fulfill the scientific value to be published.
Comments on the Quality of English LanguageMinor mistakes.
Author Response

(The authors gave the same response as above.)
